# Guided-Mode Resonance-Based Relative Humidity Sensing Employing a Planar Waveguide Structure

**DOI:** 10.3390/s20236788

**Published:** 2020-11-27

**Authors:** Petra Urbancova, Jakub Chylek, Petr Hlubina, Dusan Pudis

**Affiliations:** 1Department of Physics, Faculty of Electrical Engineering and Information Technology, University of Zilina, Univerzitna 1, 01026 Zilina, Slovakia; pudis@fyzika.uniza.sk; 2Department of Physics, Technical University Ostrava, 17. listopadu 2172/15, 708 00 Ostrava-Poruba, Czech Republic; jakub.chylek@vsb.cz (J.C.); petr.hlubina@vsb.cz (P.H.)

**Keywords:** guided-mode resonance, planar waveguide structure, humidity sensor, sensitivity, resonance wavelength, figure of merit

## Abstract

In this paper, we present a new type of guided-mode resonance (GMR)-based sensor that utilizes a planar waveguide structure (PWS). We employed a PWS with an asymmetric three-layer waveguide structure consisting of substrate/Au/photoresist. The ellipsometric characterization of the structure layers, the simulated reflectance spectra, and optical field distributions under GMR conditions showed that multiple waveguide modes can be excited in the PWS. These modes can be used for refractive index sensing, and the theoretical analysis of the designed PWS showed a sensitivity to the refractive index up to 6600 nm per refractive index unit (RIU) and a figure of merit (FOM) up to 224 RIU^−1^. In response to these promising theoretical results, the PWS was used to measure the relative humidity (RH) of moist air with a sensitivity up to 0.141 nm/%RH and a FOM reaching 3.7 × 10^−3^%RH^−1^. The results demonstrate that this highly-sensitive and hysteresis-free sensor based on GMR has the potential to be used in a wide range of applications.

## 1. Introduction

Recent progress in controlling and guiding light using a planar waveguide has been strongly motivated by the acquisition of new optical devices with unique properties [1]. Guided mode resonance (GMR) is a very promising development that is based on the excitation of guided modes in a planar waveguide under phase-matching condition using a special coupling element such as a prism or diffraction grating [2]. Some of the guided light in the waveguide can leak and under certain phase-matching conditions these leaky modes can interfere with the non-coupled reflected or transmitted light waves. This interaction causes an ultra-narrow resonance dip or peak in the reflectance or transmittance spectra at a particular wavelength, angle, and polarization [3]. Due to its controllable linewidth, the GMR effect has become an important concept in optical devices such optical sensors, optical polarizers, band-pass and band-stop filters, electro-optical switches, and modulators [2,3,4,5,6,7,8].

The resonant condition of GMR is very sensitive to changes in the refractive index in the surrounding environment, hence GMR-based structures are very good candidates for high-performance refractive index (RI) sensors [2]. Changes in the refractive index of the surrounding medium are accompanied by a shift in wavelength [9], angle [10], intensity [11] or phase [12], which can be detected by the corresponding detection scheme. The RI describes the optical parameters of a material, which are directly related to its composition, thus RI detection is an important concept in relation to sensing in biology, chemistry, and other fields [13]. GMR-based biosensors play an important role because they are label-free and able to provide real-time detection [14,15].

The structural parameters and the optical properties of the GRM structure are crucial factors that determine the performance of the GMR-based sensor [3]. Some classic planar dielectric waveguides include a metal layer in the waveguide architecture, and these can provide unique optical properties [16]. Interesting results have been found for the symmetric metal-dielectric-metal (MDM) and dielectric-metal-dielectric (DMD) geometries and also their asymmetric types [1,17,18]. In particular, metal-cladding waveguide structures have shown very interesting results in high-performance sensing applications. Zhou et al. presented absorption-based sensing by employing a symmetrical metal-cladding waveguide, in which liquid analyte served as the guiding layer [19]. Wang et al. studied and fabricated an RI sensor with high sensitivity based on an asymmetric metal-cladding dielectric waveguide structure. In this waveguide structure, an analyte was used as the guiding layer and gold and air were used for the cladding layers [13,20]. Nesterenko et al. performed an analytical study of a low-loss waveguide structure and demonstrated the degree of waveguide mode loss by controlling the thickness of the spacer layer between the absorptive and waveguide layer [17]. Yang et al. theoretically studied a metal-dielectric multilayer structure with high sensitivity due to the coupling between the surface plasmon polariton mode and multi-order waveguide modes [21]. Our goal was to theoretically and experimentally investigate a metal-cladding waveguide structure with a photoresist guiding layer and an analyte (a moist air) as an infinite dielectric cladding. This waveguide geometry shows very promising sensing properties.

In this paper, we present a highly sensitive GMR-based sensor that utilizes a planar waveguide structure (PWS) consisting of substrate/Au/photoresist layers intended for the measurement of relative humidity of moist air. The reflectance was evaluated using a transfer matrix method and showed sensitivity to the refractive index up to 6600 nm per refractive index unit (RIU) and a figure of merit (FOM) up to 224 RIU^−1^. Following the theoretical analysis, the PWS was used to measure the relative humidity (RH) of moist air. The spectral reflectance measurements were performed for *s*- and *p*- polarized waves reflected from the PWS. To attain resonance excitation of guided modes, we employed a BK7 coupling prism and the attenuated total reflection. The GMR effect of the PWS caused by the excitation of guided mode, which is manifested in the form of a narrow, well-pronounced reflectance dip, had a high sensitivity to the change in relative humidity (RH) of the moist air. The sensitivity to the RH and FOM were as high as 0.141 nm/%RH and 3.7 × 10^−3^%RH^−1^, respectively.

## 2. Structure Design and Theoretical Background

We proposed a PWS in the form of asymmetric three-layer waveguide as shown in Figure 1, in which the TE and TM guided modes can be excited if the resonant condition is fulfilled. A fused silica glass slide with a thickness of *t_s_* = 0.7 mm and the refractive index *n_s_* was used as a substrate. The metal cladding layer, which serves as a coupling layer is formed by an Au film with a thickness of *t_Au_* = 30 nm and the complex permittivity $\varepsilon_{Au}$. The waveguide layer consists of dielectric material AZ1505 photoresist with a thickness of *t_p_* ≈ 200 nm and the refractive index *n_d_*. The external medium is formed by moist air with a refractive index *n_air_*.

The first aim was to determine the spectral reflectance and sensing properties of the PWS. In the case of multilayer structures, there are several approaches for obtaining the reflection and transmission coefficient of the electromagnetic field. One of these is a transfer matrix method (TMM), which is considered as one of the most powerful methods in contemporary theoretical physics [22]. The TMM considers the multilayer system of *m* layers and the incident light in the form of the plane wave. Each layer of system j,(j=1,2,…,m) has the thickness $t_{j}$ and the optical properties are described by the wavelength-dependent complex refractive index ${\tilde{n}}_{j}=n_{j}+ik_{j}$ and complex dielectric function $\varepsilon_{j}=\varepsilon_{r}{}_{j}+i\varepsilon_{i}{}_{j}={\tilde{n}}_{j}^{2}$, respectively. The transfer matrix of the whole multilayer system is called a scattering matrix (**S**) and is a product of the interface matrices (**I**) and layer matrices (**L**) [23,24]. In the case of the proposed three-layer system (substrate/Au/photoresist) shown in Figure 1, the scattering matrix takes the form:(1)$$S=I_{01}L_{1}I_{12}L_{2}I_{23}L_{3}I_{34},$$
where **I***_ij_* is the matrix of refraction at the *ij* interface and **L***_j_* is the phase matrix describing the propagation through *j*-th layer, and they are defined as:(2)$$I_{ij}=\frac{1}{t_{ij}}\left[\begin{array}{cc} 1 & r_{ij}\\ r_{ij} & 1 \end{array}\right],$$
(3)$$L_{j}=\left[\begin{array}{cc} e^{i\beta_{j}} & 0\\ 0 & e^{i\beta_{j}} \end{array}\right].$$

Coefficients $t_{ij}\left(\lambda \right)$ and $r_{ij}\left(\lambda \right)$ are Fresnel transmission and reflection coefficients at interface *ij* and are different for *s*- and *p*-polarized light wave, see [24], and $\beta_{j}$ is the phase thickness of the layer *j* given as:(4)$$\beta_{j}\left(\lambda \right)=\frac{2\pi }{\lambda }n_{j}\left(\lambda \right)t_{j}\cos\theta_{j}=\frac{2\pi }{\lambda }t_{j}{\left[n_{j}^{2}\left(\lambda \right)-n_{0}^{2}\left(\lambda \right)\sin^{2}\theta \right]}^{\frac{1}{2}},$$
where $\theta_{j}$ is the angle of refraction in the layer *j*. Reflectance $R_{s,p}\left(\lambda \right)$ for TE (*s*) and TM (*p*) waves of the multilayered structure can be expressed by the matrix elements of the scattering matrix **S** as:(5)$$R_{s,p}\left(\lambda \right)={\left|\frac{S_{21}\left(\lambda \right)}{S_{11}\left(\lambda \right)}\right|}_{s,p}^{2}.$$

In the reflectance evaluations, the refractive index values of an external medium (moist air) were changed in the range of 1–1.005 with a step of 0.001 and the following dispersions of materials of the PWS were used. To model the response of the PWS, a BK7 coupling prism was included, whose dispersion is given by a Sellmeier formula:(6)$$n\left(\lambda \right)=\sqrt{1+\frac{a\lambda^{2}}{\lambda^{2}-d}+\frac{b\lambda^{2}}{\lambda^{2}-e}+\frac{c\lambda^{2}}{\lambda^{2}-f},}$$
where λ is the wavelength in µm and the Sellmeier coefficients are as follows: *a* = 1.03961212, *b* = 0.231792344, *c* = 1.01046945, *d* = 6.00069867 × 10^−3^ µm^2^, *e* = 2.00179144 × 10^−2^ µm^2^ and *f* = 1.03560653 × 10^2^ µm^2^. The dispersion of the substrate and AZ1505 photoresist was measured by a spectral ellipsometry method, when an ellipsometer RC2 (J. A. Woollam Co., Inc., Lincoln, NE, USA) was employed and the dispersion data fulfilled a Cauchy formula:(7)$$n\left(\lambda \right)=\frac{a}{\lambda^{2}}+\frac{b}{\lambda }+c+d\lambda +e\lambda^{2},$$
where *λ* is the wavelength in µm and the Cauchy coefficients for the substrate are *a* = 0.01411 µm^2^, *b* = −0.04034 µm, *c* = 1.549, *d* = −0.03365 µm^−1^ and *e* = 0.007592 µm^−2^, and for the photoresist they are a=0.03849 µm^2^, b=−0.1192 µm, c=2.184, d=−0.1672 µm^−1^ and e=0.05318 µm^−2^. The refractive indices of the substrate and AZ1505 photoresist given by Equation (7) are shown in Figure 2a. The dispersion of the Au layer obtained from ellipsometric measurements was described by the complex dielectric function given by the Drude–Lorentz model [25]:(8)$$\varepsilon_{Au}\left(\lambda \right)=1-\frac{1}{\lambda_{p}^{2}\left(1/\lambda^{2}+i/\gamma_{p}\lambda \right)}-\overset{2}{\underset{j=1}{\sum }}\frac{A_{j}}{\lambda_{j}^{2}\left(1/\lambda^{2}-1/\lambda_{j}^{2}\right)+i\lambda_{j}^{2}/\gamma_{j}\lambda },$$
where *λ* is the wavelength in nm and the parameters are specified in Table 1. The real and imaginary part of the dielectric function of Au given by this model is shown in Figure 2b.

Figure 3 shows the theoretical spectral reflectances, $R_{s}\left(\lambda \right)$ and $R_{p}\left(\lambda \right)$ for given refractive index values of the analyte layer calculated for the angle of incidence *θ* = 42.2° with the following parameters: $t_{Au}=30$ nm, $t_{p}$ = 200 nm. The calculated reflectance spectra exhibit well pronounced dips with a constant width corresponding to guided modes TE_1_, TE_0_ (Figure 3a) and TM_1_ (Figure 3b), respectively. The resonance wavelength shifts toward longer wavelengths with the increasing values of the refractive index of analyte. In Figure 3 we can see that the largest resonance wavelength shift regarding the change in RI is the TE_0_ mode, and on contrary, the mode with the smallest resonance wavelength shift is the TE_1_ mode.

To describe the sensing properties of guided modes, it is necessary to determine the RI sensitivity $S_{n}$, which is defined as the change in the resonant wavelength $\delta \lambda_{r}$ with respect to the RI change δn ($S_{n}=\delta \lambda_{r}/\delta n$). The resonance wavelength of the TM_1_ and TE_0_ modes as a function of the analyte RI is shown in Figure 4a with a second-order polynomial fit, from which the RI sensitivity can be determined. The RI sensitivity $S_{n}$ shown in Figure 4b is linearly dependent and changes approximately in the range 2300–3000 nm/RIU for the TM_1_ mode, and 3900–6600 nm/RIU for the TE_0_ mode, and the FOM, which is defined as a ratio of the sensitivity and the full width at half maximum (FWHM) of the dip ($\mathrm{FOM}=S_{n}/\mathrm{FWHM}$), achieves a value of 224 RIU^−1^.

In some cases, it is advantageous to consider the interference between the modes [26], which is attained when both the polarizer and analyzer are oriented 45° with respect to the plane of incidence, and the corresponding reflectance $R_{45}\left(\lambda \right)$ is expressed as:(9)$$R_{45}\left(\lambda \right)=\frac{1}{4}\left\{R_{s}\left(\lambda \right)+R_{p}\left(\lambda \right)+2\sqrt{R_{s}\left(\lambda \right)R_{p}\left(\lambda \right)}\cos\left[\delta_{sp}\left(\lambda \right)\right]\right\},$$
where $\delta_{sp}\left(\lambda \right)$ is the phase difference between the *s*- and *p*-polarized waves. Figure 5a shows the theoretical spectral reflectance $R_{45}\left(\lambda \right)$ that correspond to the reflectances, $R_{p}\left(\lambda \right)$ and $R_{s}\left(\lambda \right)$ shown in Figure 3a,b. It is clear that the interference affects the depth of the dips. For the TM mode the depth is increased, while for the TE modes it is decreased. As demonstrated in Figure 5b, a slight shift in the resonance wavelength is present in comparison with Figure 4a, and the sensitivities are the same as those shown in Figure 4b.

A more detailed description of the GMR effect, that is, the optical field intensities ${\left|E\right|}^{2}$ in the PWS divided by ${\left|E_{0}\right|}^{2}$, where $E_{0}$ is the incident *s*-polarized or *p*-polarized electric field, are shown in Figure 6a,b, respectively. The normalized optical field intensities in the PWS shown in Figure 6a were calculated for the angle of incidence *θ* = 42.2° and a wavelength of 536.9 nm and correspond to the TE_1_ mode, and for the same angle of incidence and a wavelength of 1478.8 nm they correspond to the TE_0_ mode. While the TE_1_ mode only exhibits a 3-fold enhancement, the TE_0_ exhibits a 42-fold enhancement with very promising sensitivity. Figure 6b shows the normalized optical field intensity in the PWS for the angle of incidence *θ* = 42.2° and a wavelength of 726.3 nm and corresponds to the TM_1_ mode, which exhibits a 47-fold enhancement, however, with faster exponential decay in the analyte than for the TE_0_ mode. Thus, it is confirmed that theTE_0_ mode has higher sensitivity than the TM_1_ mode for the considered analyte.

## 3. Experimental Setup

The PWS structure under study consists of substrate/Au/photoresist layers. Fused silica glass slide serves as a substrate. The Au film was deposited on the chemically cleaned substrate, by a thermal evaporation process using a vacuum evaporator (K975X, Quorum Technologies Ltd., Laughton, East Sussex, UK). The thickness of the Au layer was measured using a film thickness monitor (10983, Quorum Technologies Ltd.) integrated in the evaporator during the evaporation process with a resolution of 0.1 nm thickness. The thin photoresist layer was deposited using an unadulterated positive photoresist AZ1505 using a spin-coating process (SPIN150, Semiconductor Production Systems Ldt., Coventry, UK). To remove the solvent from the photoresist layer, the samples were post-baked at 80 °C for 2 min. The ellipsometry measurements confirmed a homogeneous photoresist layer with a thickness 200 ± 10 nm over the 1 × 1 cm^2^ area.

The experimental setup shown in Figure 7 was used to measure the reflectance response of the PWS and the RH sensing ability in the VIS and NIR spectral ranges. We used a halogen lamp (HL-2000, Ocean Optics, Dunedin, FL, USA) as a white light source (WLS) with launching optics connected to an optical fiber (OF) with a collimating lens (CL). The collimated light beam of a 1 mm diameter was then polarized using a linear polarizer (P) (LPVIS050, Thorlabs, Newton, MA, USA) oriented 45° with respect to the plane of incidence to generate both *p*- and *s*-polarized components. The polarized light beam was coupled to the PWS using an equilateral prism made of BK7 glass (Ealing, Inc., South Natick, MA, USA) with index-matching fluid (Cargille, Cedar Grove, NJ, USA, $n_{D}$ = 1.516). The reflected light from PWS merges with a linear analyzer (A) (LPVIS050, Thorlabs) oriented 0°, 90° and 45° with respect to the plane of incidence to generate the reflectances $R_{p}\left(\lambda \right)$, $R_{s}\left(\lambda \right)$ and $R_{45}\left(\lambda \right)$ [26], respectively. The reflected light was launched into a spectrometer (USB4000, Ocean Optics) via a read optical fiber (ROF) (M15L02, Thorlabs) during the VIS measurements. In the NIR measurements, the reflected light was launched into an FT-NIR spectrometer (FT-NIR Rocket, ARCoptix S.A., Neuchatel, Switzerland) via a microscope objective and ROF (P400-2-VIS-NIR). The PWS was attached to a sensing chamber with a volume of approximately 22 mL via an O ring. To control the RH values in the sensing chamber, an electrical humidity and temperature sensor (HTS) (HTU21D, Arduino, Ivrea, Italy) connected to a controller board (Arduino UNO) was used. The adjusting system of the RH in the chamber comprises a humidifier and a two-line peristaltic pump (BT100M, Baoding Chuang Rui Precision Pump, Co., Ltd., Baoding, China). Adjusting the RH is described in detail in [26].

## 4. Results and Discussion

The GMR-based relative humidity measurements were performed at a temperature of 22.8 °C (which was kept constant to avoid temperature cross-sensitivity), while the RH in the sensing chamber varied approximately in the range of 35%RH to 85%RH. This was based on the spectral reflectance measurements for *s*- and *p*-polarized waves reflected from the PWS. The reflectance ratios that induce the GMR effect as a function of the wavelength *λ* were measured for two external angles of incidence (see Figure 7), *α* = 16.6° and α = 20.6°, respectively. The resulting reflectance spectra are shown below and show the excitation of the guided modes accompanied by well-pronounced resonance dips in the spectra. The position of the dips that determines the resonance wavelength is red-shifted when the relative humidity of moist air increases. Figure 8a shows the measured reflectance ratio $R_{s}\left(\lambda \right)$/$R_{p}\left(\lambda \right)$ as a function of the wavelength for the external angle of incidence *α* = 16.6° and the relative humidity of the air in the range of 38.2%RH to 80.1%RH. The figure shows the GMR for the TE_1_ mode accompanied by a well-pronounced dip. The resonance wavelength shift versus the RH change in the moist air is shown in Figure 8b with a second-order polynomial fit. The resonance wavelength was determined with a precision of 0.01 nm using a zero-crossing in the first derivative of the smoothed reflectance ratio. Figure 9a shows the GMR for the TE_1_ mode at the external angle of incidence *α* = 20.6°. The relative humidity of the air was changed in the range of 36.1%RH to 85.1%RH. Figure 9b shows the resonance wavelength versus the RH change in moist air with a linear fit. The nonlinear to linear change in the resonance wavelength dependence on the RH can be attributed to the attenuated optical field in the surrounding medium. When the RH decreased, no hysteresis needed to be resolved in a quick response to the RH changes indicating that the surface optical field, as shown in Figure 6 is responsible for the sensing.

Figure 10a shows the wavelength dependence of the measured reflectance ratio $R_{45}\left(\lambda \right)$/$R_{s}\left(\lambda \right)$ responsible for the TM_1_ mode excitation, for the external angle of incidence *α* = 16.6° and the relative humidity of air in a range of 39.7%RH to 84.4%RH. In Figure 10b, shows the resonance wavelength shift towards longer wavelengths as the relative humidity of the air increases with a second-order polynomial fit of the measured data. Figure 11a shows the wavelength dependence of the measured reflectance ratio $R_{45}\left(\lambda \right)$/$R_{s}\left(\lambda \right)$ for the external angle of incidence *α* = 20.6° and the relative humidity of air in the range of 35.3%RH to 85.7%RH. Figure 11b shows the resonance wavelength shift again with the second-order polynomial fit of the measured data.

Lastly, the same measurements were performed for the TE_0_ mode excited by the *s*-polarized wave. Figure 12a or Figure 13a show the measured reflectance ratio $R_{s}\left(\lambda \right)$/$R_{p}\left(\lambda \right)$ for the external angle of incidence *α* = 16.6° and *α* = 20.6°, respectively. Figure 12b shows the resonance wavelength shift measured for the relative humidity ranging from 38.8%RH to 84.6%RH when the dependence was linearly fitted. Figure 13b shows the resonance wavelength measured for the relative humidity in a range of 36%RH to 85.8%RH with a second-order polynomial fit.

To estimate the sensing properties of the PWS structure, it is necessary to determine the sensitivity to the relative humidity $S_{RH}$, which is defined as the change in the resonant wavelength $\delta \lambda_{r}$ with respect to the change in the relative humidity δRH of moist air ($S_{RH}=\delta \lambda_{r}/\delta RH$). From the measured shifts in the resonance wavelength and their respective linear polynomial dependence on the RH, as shown in previous figures, we can determine the sensitivity $S_{RH}$ of the excited guided modes to the RH. The achieved sensitivities $S_{RH}$ in a range of 35%RH to 85%RH are shown in Figure 14.

For the TE_1_ mode, the sensitivity *S_RH_* exhibits a linear dependence on the RH in the range of 0.023 to 0.042 nm/%RH for the angle of incidence of 16.6°, as shown in Figure 14a. The sensitivity *S_RH_* of the same mode for the angle of incidence *α* = 20.6° is constant with a value of 0.033 nm/%RH. The sensitivities *S_RH_* of the TM_1_ mode for the angles of incidence *α* = 16.6° and *α* = 20.6° are shown in Figure 14b with values in the range of 0.036 to 0.063 nm/%RH and 0.037 to 0.068 nm/%RH, respectively. Finally, the highest RH sensitivity exhibits the TE_0_ mode. For the angle of incidence *α* = 16.6°, *S_RH_* reaches a constant value of 0.103 nm/%RH and for the angle of incidence *α* = 20.6°, the dependence on the RH is linear and $S_{RH}$ varies in the range of 0.085 to 0.137 nm/%RH, as shown in Figure 14a.

The highest FOM, which is defined as a ratio of the sensitivity and the FWHM of the dip ($\mathrm{FOM}=S_{RH}/\mathrm{FWHM})$, corresponds to the TE_0_ mode, which has the highest sensitivity $S_{RH}$ and the narrower resonance dip. The FOM attains a value of 3.1 × 10^−3^%RH^−1^ for the angle of incidence *α* = 16.6°, as it is evident from Figure 12, and for the angle of incidence *α* = 20.6°, the FOM is as high as 3.7 × 10^−3^%RH^−1^, as is shown in Figure 13.

Table 2 summarizes various optical RH sensors with different principles and parameters, such as RH range and sensitivity, and the proposed RH sensor outperforms a number of them [26,27,28] in terms of sensitivity. These include sensors based on surface plasmon resonance [26], surface Bloch resonance [26], whispering gallery mode resonance [27] and guided mode resonance [28]. Some of the RH sensors [29,30,31] based on photonic crystal mode resonance [29] and lossy mode resonance [30,31] have higher sensitivity. However, to achieve a substantially higher sensitivity, fiber-optic RH sensors need to be implemented [32,33]. 

## 5. Conclusions

In this paper, we employed a three-layer PWS consisting of substrate/Au/photoresist to achieve highly sensitive and hysteresis-free measurement of the relative humidity of moist air. The measurement was based on resolving GMR for *s*- and *p*-polarized waves reflected from the PWS. The GMR effect in the PWS was caused by the excitation of TE_1_, TM_1_ and TE_0_ guided modes, leading to narrow and well-pronounced dips in the reflectance spectra and a high sensitivity to the change in relative humidity of the moist air. We analyzed the sensitivity of all of the excited guided modes, and the sensitivity to the RH and FOM were as high as 0.141 nm/%RH and 188%RH^−1^, respectively.

Finally, this simple sensing structure has a number of advantages, including a high sensitivity to the RI and FOM that achieved 6600 nm/RIU and 224 RIU^−1^, respectively. In addition, there is the potential to adjust the sensitivity as a constant by choosing a suitable angle of incidence. The use of the sensor can be extended to liquid analytes (working at different angles of incidence than for a moist air) where the polymer layer is substituted by a dielectric layer such as SiO_2_ [26]. Moreover, the sensor can be operated in aggressive environments because the layer acts as a protective overlayer for Au thin film, and the fiber-optic realization is possible because the sensor can be operated in the telecommunication window, near a wavelength of 1550 nm.

Thus, the GMR-based sensor has the potential to be used in a wide range of applications.

## Figures and Tables

**Figure 1 sensors-20-06788-f001:**
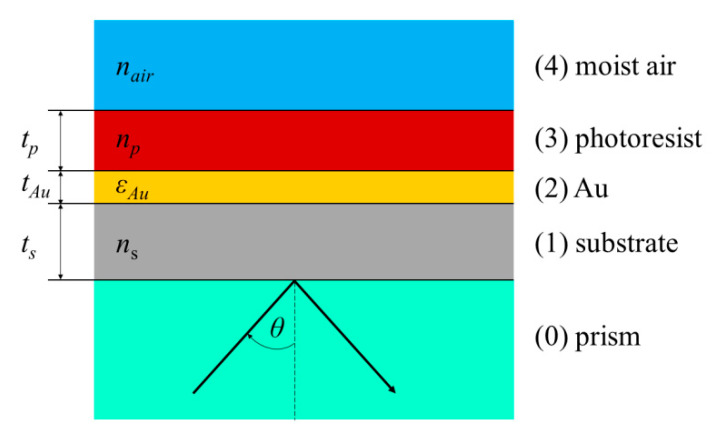
Scheme of the planar waveguide structure (PWS).

**Figure 2 sensors-20-06788-f002:**
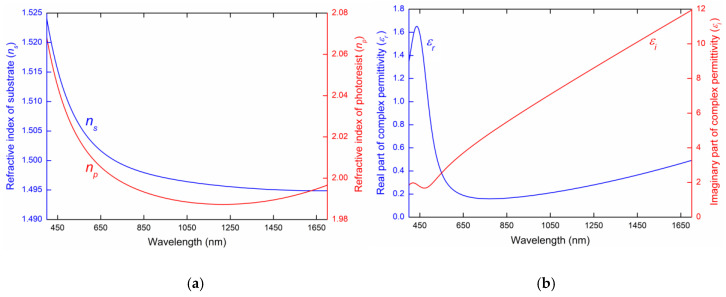
Dispersion of used materials. The refractive index of the substrate (blue line) and AZ1505 photoresist (red line) given by the Cauchy formula (**a**) and the real part (blue line) and imaginary part (red line) of the complex permittivity of the Au layer given by the Drude–Lorentz model (**b**).

**Figure 3 sensors-20-06788-f003:**
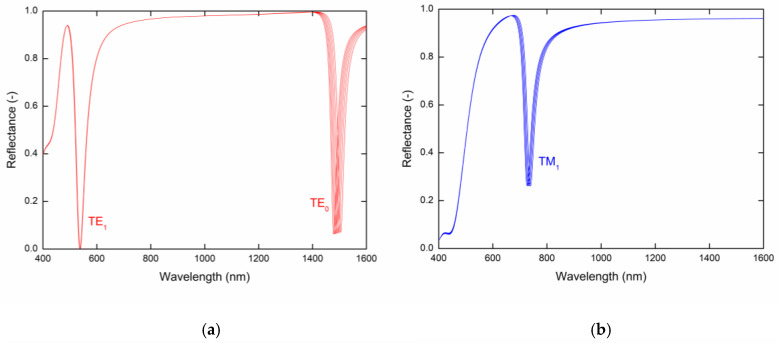
Theoretical spectral reflectances $R_{s}\left(\lambda \right)$ (**a**) and $R_{p}\left(\lambda \right)$ (**b**) for the PWS and analyte refractive index values in the range 1–1.005 with a step of 0.001.

**Figure 4 sensors-20-06788-f004:**
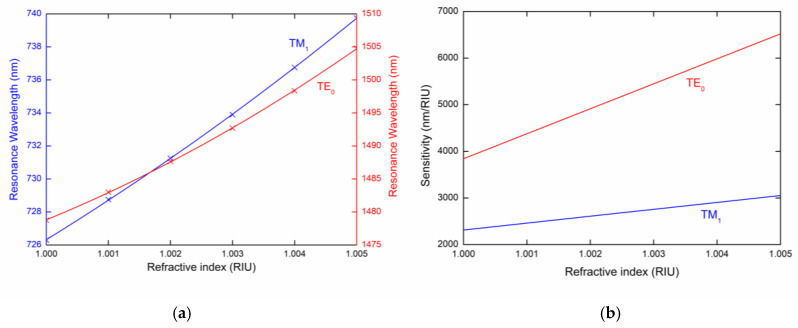
(**a**) Resonance wavelength and (**b**) sensitivity of the TM_1_ and TE_0_ modes as a function of the refractive index of the external medium.

**Figure 5 sensors-20-06788-f005:**
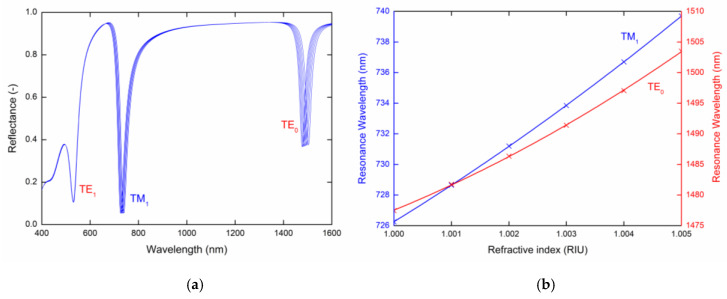
Theoretical spectral reflectances $R_{45}\left(\lambda \right)$ (**a**) and resonance wavelength as a function of the refractive index of the external medium (**b**).

**Figure 6 sensors-20-06788-f006:**
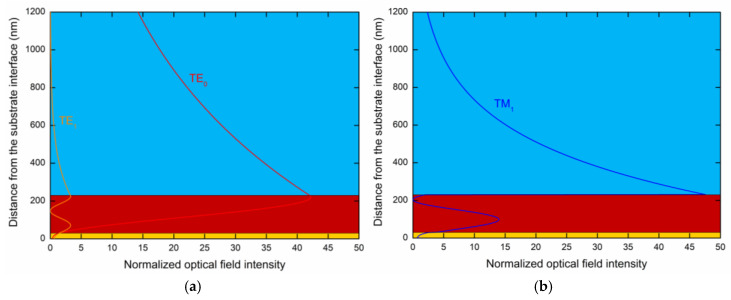
The normalized optical field intensity distribution in the PWS for the angle of incidence *θ* = 42.2° and for the wavelength (**a**) with λ= 536.9 nm exciting the TE_1_ mode and λ= 1478.8 nm exciting the TE_0_ mode, and (**b**) λ= 726.3 nm exciting the TM_1_ mode.

**Figure 7 sensors-20-06788-f007:**
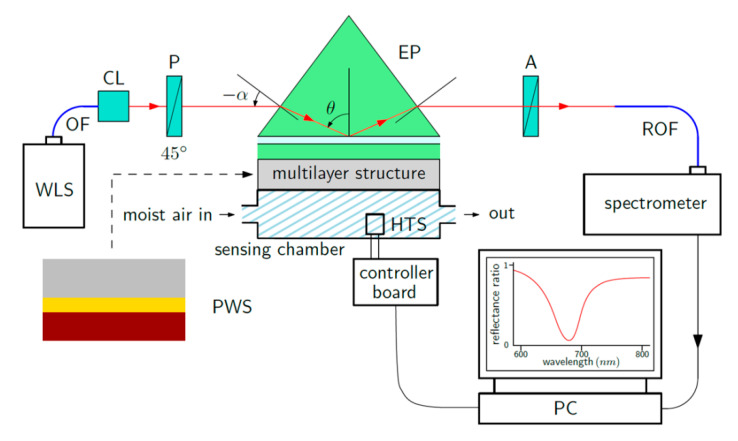
Scheme for the experimental setup.

**Figure 8 sensors-20-06788-f008:**
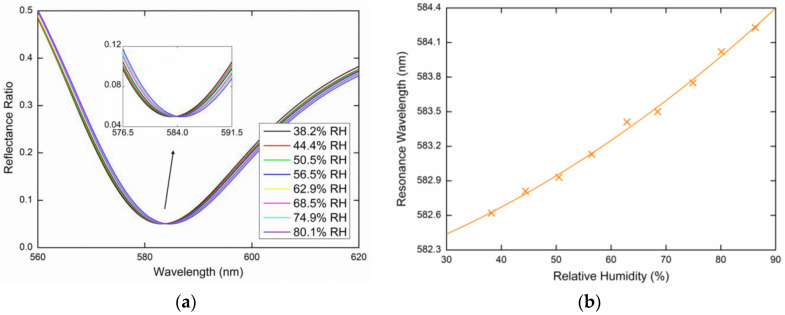
Measured spectral reflectance ratio $R_{s}\left(\lambda \right)$/$R_{p}\left(\lambda \right)$ for the external angle of incidence *α* = 16.6° (**a**), the resonance wavelength as a function of the relative humidity (RH) with a second-order polynomial fit (**b**).

**Figure 9 sensors-20-06788-f009:**
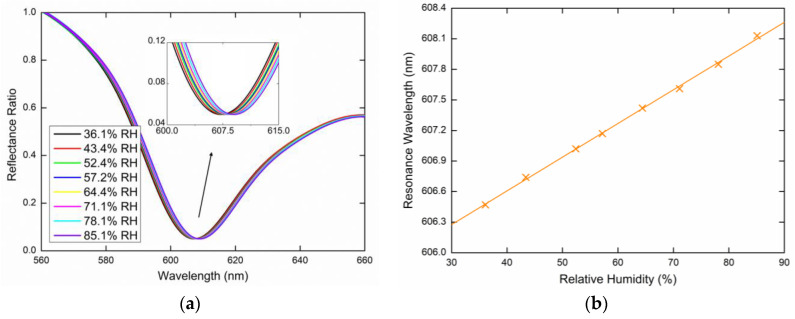
Measured spectral reflectance ratio $R_{s}\left(\lambda \right)$/$R_{p}\left(\lambda \right)$ for the external angle of incidence *α* = 20.6° (**a**), the resonance wavelength as a function of the RH with a linear fit (**b**).

**Figure 10 sensors-20-06788-f010:**
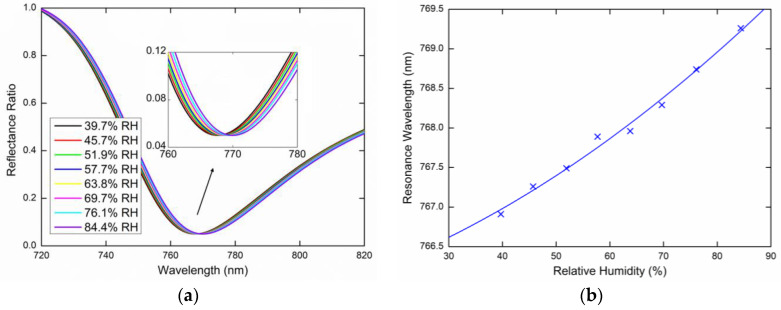
Measured spectral reflectance ratio $R_{45}\left(\lambda \right)$/$R_{s}\left(\lambda \right)$ for the external angle of incidence *α* = 16.6° (**a**), the resonance wavelength as a function of the RH with a second-order polynomial fit (**b**).

**Figure 11 sensors-20-06788-f011:**
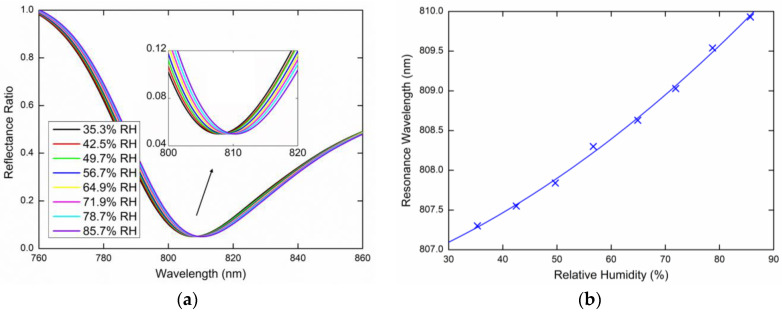
Measured spectral reflectance ratio $R_{45}\left(\lambda \right)$/$R_{s}\left(\lambda \right)$ for the external angle of incidence *α* = 20.6° (**a**), the resonance wavelength as a function of the RH with a second-order polynomial fit (**b**).

**Figure 12 sensors-20-06788-f012:**
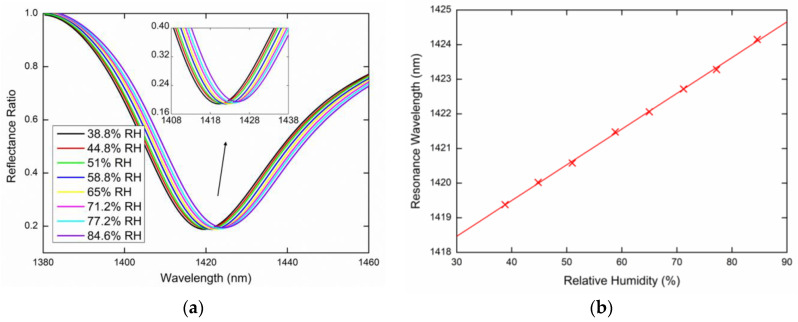
Measured spectral reflectance ratio $R_{s}\left(\lambda \right)$/$R_{p}\left(\lambda \right)$ for the external angle of incidence *α* = 16.6° (**a**), the resonance wavelength as a function of the RH with a linear fit (**b**).

**Figure 13 sensors-20-06788-f013:**
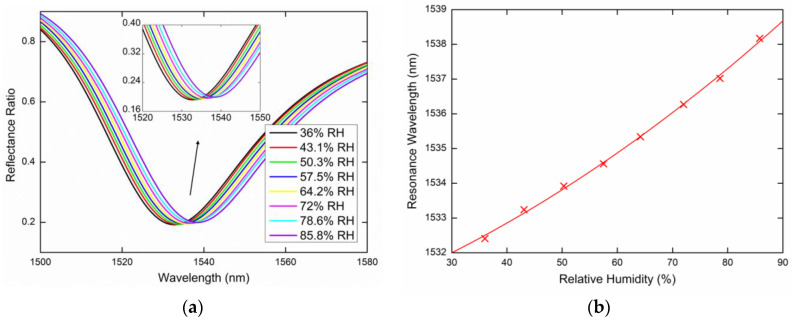
Measured spectral reflectance ratio $R_{s}\left(\lambda \right)$/$R_{p}\left(\lambda \right)$ for the external angle of incidence *α* = 20.6° (**a**), the resonance wavelength as a function of the RH with a second-order polynomial fit (**b**).

**Figure 14 sensors-20-06788-f014:**
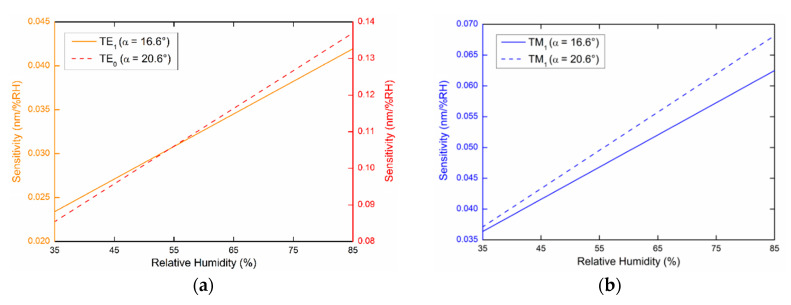
Sensitivities of guided modes measured at external angles of incidence *α* = 16.6° (**a**) and *α* = 20.6° (**b**).

**Table 1 sensors-20-06788-t001:** Parameters of the dielectric function of Au.

Drude Term Parameter	Value	Oscillator 1 Parameter	Value	Oscillator 2 Parameter	Value
$\varepsilon_{\infty }$	1	*A* _1_	1.860	*A* _2_	3.439
*λ_p_* (nm)	138.831	*λ*_1_ (nm)	419.828	*λ*_2_ (nm)	294.231
*γ_p_* (nm)	21,687.402	*γ*_1_ (nm)	−39.047	*γ*_2_ (nm)	−4192.008

**Table 2 sensors-20-06788-t002:** Optical RH sensors with different parameters.

Material	Method	RH Range	Sensitivity (nm/%RH)	Ref.
plasmonic multilayer	surface plasmon wave resonance	20–80%	0.072	[26]
dielectric multilayer	surface Bloch wave resonance	22–80%	0.065	[26]
polymer coating	whispering gallery mode resonance	0–60%	0.013	[27]
agarose gel	guided mode resonance	20–80%	0.150	[28]
porous thin film	photonic crystal mode resonance	11–84%	0.296	[29]
indium tin oxide	lossy mode resonance	65–90%	0.212	[30]
copper oxide	lossy mode resonance	30–90%	0.636	[31]

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
