# Peer review of "Guided-Mode Resonance-Based Relative Humidity Sensing Employing a Planar Waveguide Structure"

_sensors, 2020, doi:10.3390/s20236788_

Round 1

Reviewer 1 Report

The manuscript presented a study on a guided-mode resonance sensor based on a substrate/Au/photoresist planar waveguide structure. Both simulation and experimental results were presented in the paper. The humidity sensing using guided-mode resonance is novel to some extent, and a photoresist based planar waveguide structure used for sensing is interesting. The paper also shows clear structure, reasonable layout and good writing. Therefore, I recommend the manuscript for publication in Sensors after the authors address the following comments.

(1) In the fourth part “Experimental results and discussion”, the authors used “reflectance ratios inducing the GMR effect as a function of the wavelength” (line 191) for relative humidity measurements, such as Rs/Rp for TE modes and R45/Rs for TM mode. However, in the simulation part, the authors only used reflectance Rs or Rp to describe the sensing properties. Why are theoretical simulations inconsistent with experimental measurements? The authors should provide detailed explanations on this issue.

(2) In the third part “Experimental setup”, the explanation on the preparation of PWS is missing and should be supplemented for the reader who may be interested.

(3) At the end of fourth part, the authors compared proposed RH sensor with some available RH sensors. I advise that the authors add a table that lists some data (such as the structure and performance parameters) of proposed and available RH sensors for comparison, which can provide more powerful evidence for good characteristics of proposed sensor.

Author Response

List of changes

We very much appreciate inspiring questions, comments, and suggestions of reviewer. We have carefully included the comments in the text and tried to answer all the questions. We consider the reviewer´s questions to be very factual and helpful for increasing the level of the manuscript. All the changes and improvements in the manuscript are presented in the list of changes and highlighted by red color in the corrected manuscript.

Reviewer's comments:

Reviewer #1: The manuscript presented a study on a guided-mode resonance sensor based on a substrate/Au/photoresist planar waveguide structure. Both simulation and experimental results were presented in the paper. The humidity sensing using guided-mode resonance is novel to some extent, and a photoresist based planar waveguide structure used for sensing is interesting. The paper also shows clear structure, reasonable layout and good writing. Therefore, I recommend the manuscript for publication in Sensors after the authors address the following comments.

(1) In the fourth part “Experimental results and discussion”, the authors used “reflectance ratios inducing the GMR effect as a function of the wavelength” (line 191) for relative humidity measurements, such as Rs/Rp for TE modes and R45/Rs for TM mode. However, in the simulation part, the authors only used reflectance Rs or Rp to describe the sensing properties. Why are theoretical simulations inconsistent with experimental measurements? The authors should provide detailed explanations on this issue.

We included into manuscript the theoretical spectral reflectances  corresponding to the reflectances  and  shown in Figures 3a and 3b. From that new graph in Figure 5a, we can see the change in the depth of the dips and slight change in the position of the resonant wavelength. Regarding to theoretical calculation, the GMR dips corresponding to the TE and TM modes can be obtained for the reflectances Rs or Rp, or equivalently for the reflectance ratios Rs/Rp and R45/Rs (see ref. [26] in the manuscript). During simulations, the reflectance ratios are used only for the resolving the maximal depth of the dips, while in the measurement for the GMR dip acquisition, we need the reference signal. For this reason, we work with the reflectance rations Rs/Rp and R45/Rs in the experimental part.

(2) In the third part “Experimental setup”, the explanation on the preparation of PWS is missing and should be supplemented for the reader who may be interested.

The more detailed description of the fabrication process of the PWS was included into the manuscript in the beginning of “Experimental” part.

(3) At the end of fourth part, the authors compared proposed RH sensor with some available RH sensors. I advise that the authors add a table that lists some data (such as the structure and performance parameters) of proposed and available RH sensors for comparison, which can provide more powerful evidence for good characteristics of proposed sensor.

We included into manuscript table with the comparison of available optical RH sensors and their parameters – material, sensing principle, RH range and sensitivity.

Reviewer 2 Report

The manuscript with ID “sensors-1004317” and entitled “Guided-mode resonance-based relative humidity sensing employing a planar waveguide structure” is an interesting work. However, it needs some improvements before being published in the journal “Sensors”.

  1. The photoresist is not a stable material and can be affected by light illumination and moisture. The system needs an aging test.
  2. The humidity and the refractive index's measurement by inducing GMR with Kretschmann configuration can be strongly affected by temperature? The authors need to provide information about this.
  3. There are many similar reference works based on GMR. What is the novelty of this research?

Author Response

List of changes

We very much appreciate inspiring questions, comments, and suggestions of reviewer. We have carefully included the comments in the text and tried to answer all the questions. We consider the reviewer´s questions to be very factual and helpful for increasing the level of the manuscript. All the changes and improvements in the manuscript are presented in the list of changes and highlighted by red color in the corrected manuscript.

Reviewers' comments:

Reviewer #2: The manuscript with ID “sensors-1004317” and entitled “Guided-mode resonance-based relative humidity sensing employing a planar waveguide structure” is an interesting work. However, it needs some improvements before being published in the journal “Sensors”.

  1. The photoresist is not a stable material and can be affected by light illumination and moisture. The system needs an aging test.

We agree, that the photoresist AZ1505 is not a stable material, but as we observed it was not affected by light illumination and it is resistant to the moisture. We made several measurements during which we have not noticed the degradation of the structure or degradation of the sensing properties. For our application, the photoresist has sufficient chemical and physical stability. As the manufacturer declares, the photoresist can be chemically damaged by the oxidizing agents, strong acids, and bases [Datasheet_1] and [Datasheet_2]. Therefore, for the use of our type of PWS structure in a more aggressive environment, it is necessary to replace the photoresist layer by the dielectric layer such as SiO2 as we mentioned in conclusion of the manuscript.

[Datasheet_1] https://in.bgu.ac.il/en/nano-fab/Site%20Assets/Lists/MSDS%20list/AllItems/AZ%201505%20photoresist.pdf

[Datasheet_2] https://www.nanofab.utah.edu/wp-content/uploads/2018/09/AZ-1505-Photoresist-AZ-Electronic-Materials-09Apr13_v33.pdf

  1. The humidity and the refractive index's measurement by inducing GMR with Kretschmann configuration can be strongly affected by temperature? The authors need to provide information about this.

The temperature is kept constant (t = 22.8 °C) to avoid temperature cross-sensitivity. It was included in the paper.

  1. There are many similar reference works based on GMR. What is the novelty of this research?

The major available GMR sensors are grating-based and for the GMR modes excitation uses diffraction grating. We present a new concept of GMR sensor based on planar waveguide structure with high sensitivity. There has been presented several results of similar structures employing planar waveguides, but with different types of architecture, in which the analyte serves as the guiding layers. Some results were only theoretical. We prepared the GMR structure, in which the analyte forms an infinite dielectric cladding. As we shown, the enhanced optical field intensity in the proposed PWS structure leads to the high sensitivity to the refractive index change of the analyte. For this reason, the structure was used as the sensor of relative humidity and we outperform some of the available RH sensor based on other sensing methods presented in added Table 2.

Reviewer 3 Report

The paper reports theoretical and experimental investigation of a multilayer waveguide for relative humidity sensing. The work has some value and the authors must address the following comments before the paper can be considered for publication.  

  • Please mention the resolution of the spectrometer to confirm that the observed sub-nanometer shifts in the resonance wavelength are due to the changes in the relative humidity only.
  • The authors should discuss in the paper why in some cases the resonance shift vs %RH is nonlinear and in some other case linear.
  • Discussion about the fabrication details for the multilayer waveguide is missing in the manuscript.
  • Please include a SEM image/optical image of the fabricated waveguide

Author Response

List of changes

We very much appreciate inspiring questions, comments, and suggestions of reviewer. We have carefully included the comments in the text and tried to answer all the questions. We consider the reviewer´s questions to be very factual and helpful for increasing the level of the manuscript. All the changes and improvements in the manuscript are presented in the list of changes and highlighted by red color in the corrected manuscript.

Reviewers' comments:

Reviewer #3: The paper reports theoretical and experimental investigation of a multilayer waveguide for relative humidity sensing. The work has some value and the authors must address the following comments before the paper can be considered for publication.  

  • Please mention the resolution of the spectrometer to confirm that the observed sub-nanometer shifts in the resonance wavelength are due to the changes in the relative humidity only.

The resonance wavelength was determined with a precision of 0.01 nm using a zero-crossing in the first derivative of the smoothed reflectance ratio. It was included in the paper.

  • The authors should discuss in the paper why in some cases the resonance shift vs %RH is nonlinear and in some other case linear.

Nonlinear to linear change of the resonance wavelength dependence on the RH can be attributed to attenuated optical field in the surrounding medium. It was included in the paper.

  • Discussion about the fabrication details for the multilayer waveguide is missing in the manuscript.

The fabrication details for the PWS structure was included into the manuscript in the “Experimental” part.

  • Please include a SEM image/optical image of the fabricated waveguide

The prepared GMR structure is planar and homogeneous, so the SEM image/optical microscope image of the PWS does not bring any new information. The surface of the photoresist layer is subwavelength smooth with very good thickness homogeneity, what was confirmed by ellipsometry measurements. We added the explanation into the “Experimental” part of manuscript.

Round 2

Reviewer 2 Report

The manuscript with ID “sensors-1004317” and entitled “Guided-mode resonance based relative humidity sensing employing a planar waveguide structure” is an interesting work. However, I still have concerns related to the stability of the photoresist. The system may not reach the original setup after long-term use or storage of the GMR sensor. But I think it can be reserved for the readers. This report can be published as it is in the revised version.